# Self-guided versus facilitator-guided debriefing in immersive virtual reality simulation: Protocol for a randomized controlled non-inferiority trial assessing teamwork skills in medical students

Amalie Middelboe Sohlin[1]*, Anja Poulsen[1], Ida Madeline Hoffmann[1],
Line Klingen Gjærde[1,2], Stine Lund[3,4], Gritt Overbeck[5], Lone Paulsen[6], Todd P. Chang[7],
Joy Yeonjoo Lee[8], Jette Led Sørensen[2,4], Jesper Kjærgaard[1]

1 Department of Paediatrics and Adolescent Medicine, Copenhagen University Hospital – Rigshospitalet, Copenhagen, Denmark, 2 Mary Elizabeth's Hospital and Juliane Marie Centre, Copenhagen University Hospital – Rigshospitalet, Copenhagen, Denmark, 3 Department of Pediatrics, Copenhagen University Hospital North Zealand, Hillerød, Denmark, 4 Department of Clinical Medicine, Faculty of Health and Medical Sciences, University of Copenhagen, Copenhagen, Denmark, 5 Department of Public Health, Faculty of Health and Medical Sciences, University of Copenhagen, Copenhagen, Denmark, 6 H.C. Andersen Children's Hospital, Odense University Hospital, Odense, Denmark, 7 Children's Hospital Los Angeles, Keck School of Medicine of University of Southern California, Los Angeles, California, United States of America, 8 Faculty of Governance and Global Affairs, The Hague, Leiden University, Netherlands

* amalie.middelboe.andersen@regionh.dk

## Abstract

Simulation-based medical education has been shown to be more effective but also logistically demanding and costly compared to other educational strategies in developing medical skills. Immersive virtual reality is an emerging technology enabling learners to train without a facilitator through computer-generated feedback, offering the potential for increased flexibility in the timing and location of the training and reduced costs. However, little is known about whether immersive virtual reality simulation yields similar results with and without a facilitator. The aim of this study is to compare the effects of self-guided compared to facilitator-guided debriefing for immersive virtual reality simulation-based pediatric emergency team training. We will conduct a randomized, controlled, single-blinded non-inferiority study with a parallel group, pretest-post-test design. 88 medical students (44 teams) will be randomized to undergo immersive virtual reality simulation-based pediatric emergency team training with either self-guided or facilitator-guided debriefing. We will assess the teams before and after the virtual reality intervention in a mannequin-based simulation. The mannequin-based simulation will be videorecorded, and two independent raters, blinded to group allocation, will assess the recordings using validated scales measuring teamwork skills (primary outcome), ABCDE adherence, and time to critical actions. We will further collect data on perceptions of debriefing quality, motivation, workload, usability, and cybersickness. To account for repeated measures and clustering within teams, we will apply a linear mixed model for data analysis. This study

**Data availability statement:** No datasets were generated or analyzed during the current study. Deidentified research data from this study will, due to data security considerations, be available upon request from the corresponding author when the study is completed and published.

**Funding:** J.K. received a grant for this work from Gangstedfonden, grant number A43424, https://www.gangstedfonden.dk/ and Laerdal Foundation, grant number 2024-0321, https://laerdalfoundation.org/ . A.P. received a grant for this work from Helsefonden, grant number 21-B-385, https://helsefonden.dk/ . The funders did not play any role in the study design, data collection and analysis, decision to publish, or preparation of the manuscript.

**Competing interests:** The authors have declared that no competing interests exist.

**Abbreviations:** ABCDE, airways, breathing, circulation, disability, and exposure; CI, confidence intervals; CTS, clinical teamwork scale; DASH-SV, debriefing assessment for simulation in healthcare student version; IMI, intrinsic motivation inventory; SUS, system usability scale; NASA-TLX, national aeronautics and space administration task load index; RQ, research question; VR, virtual reality; VRSQ, virtual reality sickness questionnaire.

aims to provide insight into the effects of self-guided versus facilitator-guided debriefing in immersive virtual reality simulation, with implications for the future development and implementation of immersive virtual reality simulation in medical education.

We have registered the trial on ClinicalTrials.gov (identifier: NCT06956833).

## Introduction

Simulation-based medical education has been shown to significantly enhance the skills of healthcare professionals and students, as well as improve patient outcomes [1–6]. Several systematic reviews and meta-analyses have demonstrated that simulation-based medical education outperforms traditional clinical education and other educational strategies in improving a wide range of medical skills [4,5]. However, simulation-based medical education is also more logistically demanding and costly compared to many other educational strategies [5], prompting calls for research into strategies that maximize its benefits while increasing access and minimizing costs [7].

Immersive virtual reality (VR) is an emerging technology within simulation-based medical education [8–11]. Recent research suggests that VR can provide learning outcomes comparable to other simulation modalities, while potentially offering greater flexibility in training schedules and reducing costs [9,11]. One of its main advantages in enhancing flexible use and reducing costs is the potential reduction in educational professionals needed to facilitate the simulations, as VR scenarios can be programmed to adapt dynamically to learners' actions, allowing learners to train independently of a facilitator [12]. However, little is known about whether the presence of a facilitator significantly influences the effectiveness of VR simulation.

In simulation-based medical education, the facilitator plays a key role in guiding the debriefing conversation [13]. Debriefing, described as a structured reflection aiming to explore and learn from the simulation-based experience [14,15], has been shown to significantly enhance clinical performance outcomes [16–20]. While facilitator-guided debriefing is widely regarded as the gold standard [13], learners can also lead their own reflection through *self-guided debriefing* [14]. In such cases, the debriefing is typically supported by cognitive aids, such as recordings of the simulation or a debriefing framework, to support self- and peer-assessment [14,21].

The self-guided approaches are grounded in theories of self-regulated learning, such as self-determination theory, which suggest that autonomy may enhance learner engagement and motivation [22]. Research suggests that self-guided approaches may offer additional benefits, including increased ownership and commitment to change [23], flexibility in the duration of the training [24,25], and greater motivation for self-regulated learning [26]. However, learners with lower self-regulation skills may struggle to evaluate their own performance, perceive the process as less effective, be less motivated, and experience increased cognitive load [25,27,28].

While self-guided and facilitator-guided debriefing have been compared in other domains of simulation-based medical education, such as mannequin- and

screen-based simulation [18,21,29], little is known about the effectiveness of self-guided debriefing in VR [30]. For VR to fulfil its potential as an independent and scalable simulation-based medical education modality, it is essential to determine whether self-guided debriefing can match the effectiveness facilitator-guided debriefing in a VR context.

### Previous studies on immersive VR in simulation-based emergency training

Little is known about the effect of immersive VR on teamwork skills. This represents a significant gap in the literature, as effective teamwork and communication are critical to preventing patient harm [31–33]. In the field of emergency training, VR has shown promising results for improving the individual skills of healthcare professionals and students, such as clinical reasoning and task prioritization [34–38], triage in mass causality incidents [39], memorization of the ABCDE approach [40], and neonatal resuscitation skills [41]. However, there has been a recent call for research on the effect of VR on *team-based* skill performance and process measures [10]. Further, while one of the proposed key advantages of VR is the possibility of facilitator-independent training, little is known about the role of the facilitator and different debriefing methods in VR for emergency training [10].

### Pilot study

We conducted a randomized, controlled, single-blinded pilot study with parallel group, pretest-posttest design from September to December 2023 [30]. We randomized 24 healthcare professionals (trainee doctors and nurses) from four pediatric units at Copenhagen University Hospital, Rigshospitalet, to participate in VR simulation-based pediatric emergency team training with either self-guided or facilitator-guided debriefing. Teams were assessed at baseline and at one month follow-up in a mannequin-based pediatric emergency simulation. Two independent raters blinded to group allocation assessed team performance based on video-recordings of the mannequin-based simulations. Participants completed questionnaires on their perceptions of debriefing quality, motivation, usability, and workload.

Both interventions were found acceptable and usable. However, nearly half of the participants reported some degree of cybersickness, prompting recommendations to reduce VR exposure, ensure breaks between immersions, and optimize headset alignment. Additionally, coordinating clinically working healthcare professionals across four pediatric units to participate at across three time points proved logistically challenging. To support recruitment feasibility and reduce loss to follow-up in a definitive trial, we decided to limit data collection time points and include participants with more flexible schedules, such as students.

Preliminary results showed significant improvements from baseline to follow-up in both groups for teamwork skills (facilitator-guided mean 1.2, confidence interval (CI) 0.5 to 1.9, p = 0.005; self-guided mean 1.4, CI 0.8 to 2.0, p < 0.001) and ABCDE adherence (facilitator-guided mean 0.5, CI 0.1 to 1.0, p = 0.02; self-guided mean 0.5, CI 0.1 to 0.9, p = 0.01). The facilitator-guided group rated the debriefing quality higher (mean difference 1.4, CI 0.1 to 2.7, p = 0.04), but usability, workload, and motivation were similar across debriefing groups. We concluded that a study designed for testing a non-inferiority hypothesis can be relevant for conclusive evidence.

### Aims and research questions

The aim of this study is to evaluate the effects of immersive VR simulation-based pediatric emergency team training with self-guided versus facilitator-guided debriefing in a randomized, controlled, non-inferiority study with parallel-group, pretest-posttest design.

For the primary outcome, we will investigate the following research question:

**Research question 1 (RQ1 – effect on teamwork skills)**

Does VR-based pediatric emergency team training with self-guided debriefing yield a non-inferior effect on medical students' non-technical team performance (teamwork skills) compared to VR with facilitator-guided debriefing?

For the secondary outcomes, we will investigate the following research questions:

**Research question 2 (RQ2 – effect on overall team performance)**

Do the two interventions result in a similar effect on medical students' technical team performance (ABCDE adherence and time to critical actions)? (RQ2a)

Does VR improve overall team performance (teamwork skills, ABCDE adherence, and time to critical actions) in both intervention groups? (RQ2b)

**Research question 3 (RQ3 – perceived effectiveness and engagement)**

Do the two interventions result in a similar engagement in VR simulation and debriefing (intrinsic motivation and perceived debriefing quality)?

**Research question 4 (RQ4 – cognitive load)**

Do the two interventions result in a similar cognitive load?

**Research question 5 (RQ5 – usability and cybersickness)**

What is the incidence of cybersickness and the perceived usability of VR in the two interventions?

## Methods

### Setting and population

The study will be conducted at Copenhagen University Hospital, Rigshospitalet, and Odense University Hospital, Denmark. Eligible participants will be medical students enrolled at Faculty of Health and Medical Sciences, Copenhagen University and University of Southern Denmark, who are within two years of graduation. Exclusion criteria will be lack of informed consent.

Participants will be recruited between May and October 2025 via emails sent to the consultants responsible for medical students' clinical placements at pediatric departments in the Capital Region of Denmark, Region Zealand, and the Region of Southern Denmark. These consultants will be encouraged to inform medical students, both in person and via email, about the opportunity to participate in the study. Additionally, information and invitations to participate will be shared on social media platforms targeting medical students in the relevant regions. Written informed consent, including permission to record and store all video data for the research project, will be obtained from all participants upon enrolment.

### Study design

We will conduct a single-blinded, randomized controlled non-inferiority study with parallel-group, pretest-posttest design, following the Standard Protocol Items: Recommendations for Interventional Trials (SPIRIT) statement on clinical trial protocols [42].

Teams of medical students will be randomized at a 1:1 allocation ratio to go through immersive VR simulation-based pediatric emergency team training with self-guided or facilitator-guided debriefing. To evaluate the effect of the interventions on team performance, the teams will be assessed managing a mannequin-based pediatric emergency before and after the intervention (Fig 1). The mannequin-based simulation will be video recorded, and two independent observers blinded to group allocation will rate the collective performance of each team based on the videos.

### Interventions

Both interventions will last four hours, including one hour of immersive VR simulation and one hour of debriefing (Fig 2). To minimize cybersickness, participants will spend at least 30 minutes outside the VR environment between scenarios. If cybersickness symptoms occur, they will be prompted to exit the simulation and continue by observing on a screen, providing supervision to their teammate.

In the facilitator-guided group, a medical doctor with formal training and experience in simulation and debriefing will guide the debriefings using the same VR-modified PEARLS framework as the self-guided group. This ensures that the only planned difference between groups is the presence or absence of a facilitator [30,43].

| | STUDY PERIOD | | | | |
|---|---|---|---|---|---|
| | Enrolment | Allocation | Post-randomization | | |
| TIMEPOINT* | -t1 to 0 | 0 | t1 | t2 | t3 |
| **ENROLMENT:** | | | | | |
| **Eligibility screen** | X | | | | |
| **Informed consent** | X | | | | |
| **Randomization** | | X | | | |
| **INTERVENTIONS:** | | | | | |
| *VR with self-guided debriefing* | | | | X | |
| *VR with facilitator-guided debriefing* | | | | X | |
| **ASSESSMENTS:** | | | | | |
| *Sociodemographic data* | X | | X | | |
| *CTS* | | | X | | X |
| *ABCDE* | | | X | | X |
| *Time to critical action* | | | X | | X |
| *DASH-SV* | | | | X | |
| *IMI* | | | | | X |
| *NASA-TLX* | | | | | X |
| *SUS* | | | | | X |
| *VRSQ* | | | | | X |

**Fig 1. Participant timeline: Schedule of enrolment, interventions, and assessments as recommended by the 2025 SPIRIT statement.** * Enrolment and allocation will take place 1–8 weeks prior to t1. t1, t2, and t3 will be conducted in the morning, midday, and afternoon of the same day, respectively. ABCDE: airways, breathing, circulation, disability, and exposure; CTS: clinical teamwork scale; DASH-SV: debriefing assessment for simulation in healthcare student version; IMI: intrinsic motivation inventory; NASA-TLX: national aeronautics and space administration task load index; SUS: system usability scale; VR: virtual reality; VRSQ: virtual reality sickness questionnaire.

## Development of immersive VR pediatric emergency scenarios and core game mechanics

Three immersive VR scenarios were developed and tested in our pilot study [30]. The scenarios depict three pediatric emergency cases: sepsis, respiratory distress, and anaphylaxis. Interactive elements are embedded in the virtual environment and trigger predefined changes in the patient's vitals and clinical condition. Dropdown menus are used to simulate palpating of the patient. Participants will navigate the emergency room via teleportation. The cases were developed by

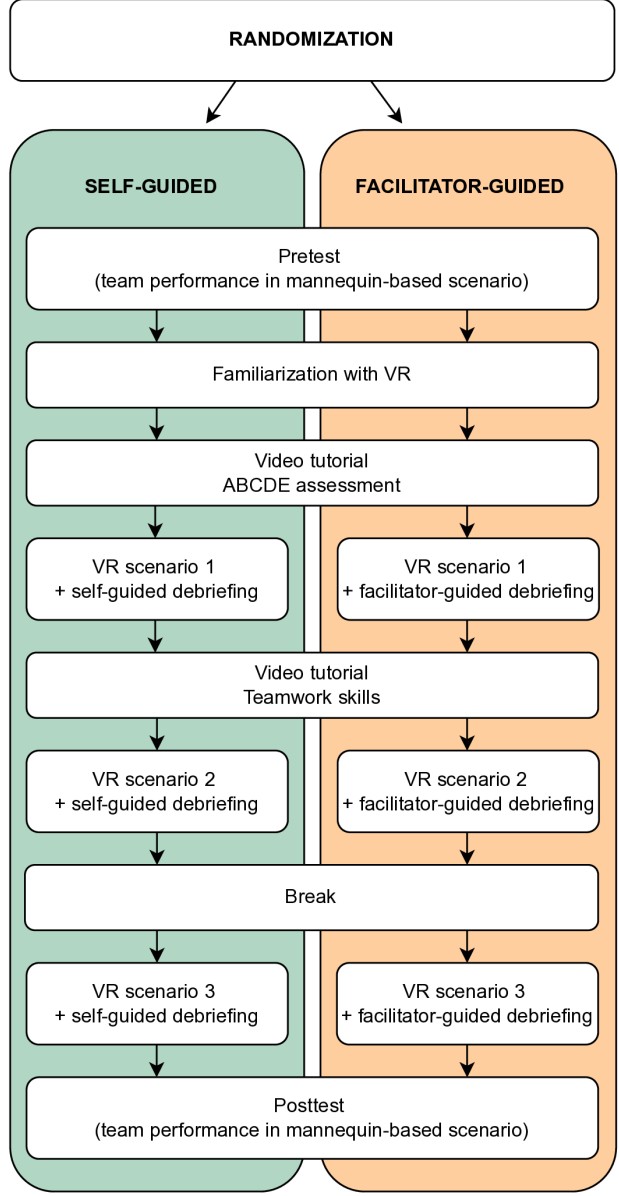

**Fig 2. Illustration of pretest, posttest, and intervention phases.** The self-guided group will receive a debriefing script based on the Promoting Excellence And Reflective Learning in Simulation (PEARLS) framework [43] adapted for VR and self-guided use [30,44]. No facilitator will be present, but participants may request technical support. A research assistant will introduce the VR simulation, administer questionnaires, and confirm that all scenarios and debriefings were completed.

A.M.S., L.P., J.K., and A.P.; validated by A.P., J.L.S., L.K.G., and S.L.; and implemented in UbiSim's© VR platform by A.M.S. and I.M.H. Scenarios will be delivered using Oculus Quest 2 headsets.

## Development of assessment scenarios

The pretest and posttest assessment scenarios depict an 8-year-old with asthma and a 3-month-old with pneumonia. Participants will perform the scenarios in a mannequin-based simulation to ensure that any improvement in team performance is not

due to familiarization with the VR simulation. To reduce potential carryover or learning effects between pre- and post-intervention assessments, distinct but comparable scenarios are used at each time point. Scenarios are matched in complexity and learning objectives but differ in content to minimize practice effects. The two assessment scenarios were tested in our pilot study and found to be similar in difficulty level [30]. A.M.S., J.K. and A.P. created the cases, with revisions from J.L.S., L.K.G., L.P., and S.L.

## Outcome measures

Outcome measures and assessment tools are further described in Table 1.

Baseline characteristics will be collected on all participants prior to the intervention, including gender, age, current semester, and prior experience in simulation-based training, computer games, and immersive VR.

**RQ1- effect on teamwork skills (primary outcome).** Two independent raters blinded to group allocation will rate teamwork skills in the pretest and posttest videos using Clinical Teamwork Scale (CTS) [45]. CTS is designed and validated to rate teamwork skills in simulated pediatric emergencies [45]. CTS has been professionally translated to Danish and demonstrated excellent interrater reliability in our pilot study [30].

**RQ2 – effect on overall team performance (secondary outcomes).** Two independent raters blinded to group allocation will rate team performance in pretest and posttest videos using the ABCDE checklist, time to critical actions, and CTS.

The ABCDE checklist is modified from the checklist by Hultin et al [46], and measures adherence to the European Pediatric Advanced Life Support guidelines on ABCDE assessment [30,52]. The checklist has demonstrated high interrater reliability in our pilot study [30].

Time to critical actions will be measured as time to each of the following predefined actions: administration of oxygen (pneumonia case) or beta2 agonist (asthma case), administration of fluid bolus 10 mL/kg body weight, and completion of ABCDE assessment.

**RQ3 – perceived effectiveness and engagement (secondary outcomes).** Participants will complete the Debriefing Assessment for Simulation in Healthcare Student Version (DASH-SV) [47] and Intrinsic Motivation Inventory (IMI) [48].

DASH-SV is a validated questionnaire designed for learners to evaluate the effectiveness of a debriefing. For the self-guided group, 'The instructor' was replaced with 'We' in consultation with the DASH-SV© developers to reflect the self-guided context, e.g., replacing "The instructor provoked in-depth discussions that led me to reflect on my performance" with "We provoked in-depth discussions that led me to reflect on my performance".

IMI is a validated measure of intrinsic motivation related to a specific activity [48]. In this study, we will use the 7-item interest/enjoyment subscale, which is considered the direct measure of intrinsic motivation. IMI and DASH-SV have previously been translated professionally to Danish and tested in a Danish context [30,53].

**RQ4 – cognitive load (secondary outcomes).** Participants will complete the validated NASA Task Load Index (NASA-TLX) [49], a tool designed to assess perceived workload. A professional Danish translation has previously been completed [30].

**RQ5 – cybersickness and usability (secondary outcomes).** Participants will report symptoms of cybersickness using the validated Virtual Reality Sickness Questionnaire (VRSQ) [51], and perceived usability will be assessed with the validated System Usability Scale (SUS) [50]. SUS has been professionally translated and validated in a Danish context [48]. At the bottom of the VRSQ, we will add a field asking participants to indicate the duration of their symptoms.

## Data collection and management

Participant recruitment will take place from May 6th, 2025, to October 31st, 2025. Data collection began on May 6th, 2025, and is expected to be completed by November 2025. Results are anticipated by June 2026. Schedule of enrolment, interventions, and assessments is presented in Fig 1.

**Table 1. Assessment tools, research hypothesis, and outcome measures.**

| | Assessment tool | Research question | Outcome measure |
|---|---|---|---|
| Team-level outcome (n = 22 teams in each arm) | **Primary outcome** | | |
| | **Clinical Teamwork Scale (CTS®)** [45]<br>− 15 teamwork skills in five main domains: overall teamwork, communication, situational awareness, decision-making<br>− 11-point Likert scale ranging from 1 = unacceptable to 11 = perfect | RQ1: Is the change from pretest to posttest in the self-guided group non-inferior compared to the facilitator-guided group, with a non-inferiority margin of 0.5?<br>RQ3: Is there a significant increase from pretest to posttest in both groups? | Mean CTS score (mean across 15 items and across the two raters.) |
| | **Secondary outcomes** | | |
| | **ABCDE checklist** [46]<br>− 20 items related to paediatric ABCDE assessment<br>- Checklist ranges from 0 = not initiated to 3 = performed consistently during the whole simulation | RQ2: Is the change from pretest to posttest in the self-guided group comparable to that in the facilitator-guided group?<br>RQ3: Is there a significant increase from pretest to posttest in both groups? | Mean ABCDE checklist score (mean across 20 items and across the two raters.) |
| | **Time to critical actions**<br>Time (in seconds) to three predefined actions. If the team does not perform the action, they will be assigned a time of 600 seconds (maximum length of scenario). | RQ2: Is the change from pretest to posttest in the self-guided group comparable to that in the facilitator-guided group?<br>RQ3: Is there a significant increase from pretest to posttest in both groups? | Mean time (mean across the three predefined critical actions and across the two raters.) |
| Individual level outcomes (n = 44 participants in each arm) | **Debriefing Assessment for Simulation in Healthcare Student Version (DASH-SV)** [47]<br>− 6 items related to perceived effectiveness of the debriefing<br>− 7-point Likert scale ranging from 1 = extremely ineffective to 7 = extremely effective | RQ4: Is the mean in the self-guided group comparable to that in the facilitator-guided group? | Mean DASH-SV score (mean across six items and across the three debriefings) |
| | **Intrinsic Motivation Inventory (IMI)** [48]<br>− 7 items from the interest/enjoyment dimension, related to participants motivation.<br>− 7-point Likert scale ranging from 1 = not at all true to 7 = very true | RQ4: Is the mean in the self-guided group comparable to that in the facilitator-guided group? | Mean IMI interest/enjoyment score (mean across seven items) |
| | **NASA Task Load Index (NASA-TLX)** [49].<br>- Six items related to participants perceived workload.<br>- Scale ranging from 0 to 21. Scores are subsequently converted into a single score from 0-100. | RQ5: Is the mean in the self-guided group comparable to that in the facilitator-guided group? | Mean NASA-TLX score (mean across six items) |
| | **System Usability Scale (SUS)** [50]<br>− 10 items related to perceived usability.<br>- Likert-scale ranging from 1 = strongly disagree, 5 = strongly agree. The scores are subsequently converted into a single SUS score from 0-100 [50]. | Exploratory – descriptive statistics | Mean SUS score (mean across ten items) |
| | **Virtual Reality Sickness Questionnaire (VRSQ)** [51]<br>− 9 symptoms of cybersickness<br>- Likert-type scale ranging from 0 = none to 3 = severe. | Exploratory – descriptive statistics | Mean VRSQ score (mean across 9 items) |

ABCDE, airways, breathing, circulation, disability, and exposure; NASA, national aeronautics and space administration task load index; RQ, research questions.

Teams will be assessed before the intervention managing a mannequin-based pediatric emergency (pretest) and again after the intervention (posttest). The mannequin-based simulation will be video recorded and two independent, blinded raters will rate team performance based on the videos. Teams will be randomly assigned one case at pretest and complete

the other at posttest. Scenarios will be recorded using a GoPro HERO 4 camera. The raters will be a medical doctor and a medical student, both trained and experienced in assessing team performance using these instruments.

Additionally, after the intervention all participants will complete questionnaires on perceived debriefing quality, motivation, workload, cybersickness, and usability (Table 1 and Fig 1).

Pretest and posttest videos will be stored on a secure hospital drive as logged files, accessible to only to authorized study personnel and are permanently deleted after completion of analyses. All other data will be stored in a secure online database (REDCap) [54] and be pseudonymized before imported into R (version 4.1.2) for statistical analysis [55].

## Sample size calculation

The sample size calculation is based on the assumption that self-guided debriefing is non-inferior to facilitator-guided debriefing for the primary outcome (teamwork skills measured by the CTS© scale.) We set a non-inferiority margin of 0.5 points on the 0–10 CTS© scale, meaning that a difference of less than 0.5 points on the CTS© scale is not expected to constitute a meaningful difference in teamwork skills. The non-inferiority margin was informed by expert consensus, our experience rating teamwork skills in the pilot study, and pilot data [30]. Based on a one-sided two-sample t-test with a non-inferiority margin of 0.5, an alpha of 0.05, a power of 80%, and a standard deviation of 0.63 derived from our pilot data [30], the required sample size is 20 teams (40 participants) in each arm. Assuming a 10% dropout rate, we plan to include approximately 44 participants in each arm, and 88 participants in total. The sample size calculation was performed in R version 4.1.2 [55].

## Randomization and blinding

Medical students will enroll in the study in teams of two based on schedule availability, without matching for prior teamwork experience, clinical rotation level, or other characteristics. Each team will be treated as a cluster and randomly allocated to the self-guided or facilitator-guided group at a 1:1 allocation ratio.

The principal investigator will generate the randomization sequence using Sealed Envelope's online randomization service [56]. Upon participant enrolment, the principal investigator will assign each team a team-ID based on the order of enrolment and allocate teams to interventions according to the randomization sequence.

The raters assessing the pretest and posttest videos will be blinded to group allocation. Due to the nature of the interventions, blinding of participants is not feasible. However, participants will not be informed of their group allocation until the intervention day, after completing the pretest, the VR scenarios used for intervention delivery are pre-programmed and identical across the intervention groups to ensure standardization, and the performance evaluation scenarios will follow standardized protocols. The researchers doing the statistical analyses will be blinded to allocation of participants.

## Statistical analysis

We will report descriptive statistics on baseline characteristics of each group, reporting continuous variables using median and interquartile range and categorical variables as n/total N (%). Descriptive statistics will be reported for all outcomes at each time point as mean, standard deviations, median, and interquartile range.

We will analyze the team-level data (RQ1 and RQ2) using a constrained linear mixed model with inherent baseline adjustment. Visit (pretest or posttest), assessment scenario (pneumonia or asthma), and the constrained interaction between visit and treatment group (self-guided vs facilitator-guided) will be included as fixed effects. An unstructured covariance pattern will be applied to account for the correlation between repeated measurements and potential changes in variance over time and account for missing data. The mean between the two raters will be calculated prior to statistical analysis.

To analyze individual-level outcomes (RQ3 and RQ4), we will apply a constrained linear mixed model with treatment group as fixed effect and a blocked compound symmetry pattern to account for intra-class correlation between individuals in the same team.

Missing data will be handled within the linear mixed model under the assumption that data are missing at random by including all available data points without requiring imputation. For missing individual-level covariate or questionnaire data, we will first assess the extent and pattern of missingness. If missingness exceeds a minimal threshold of >5%, we will apply multiple imputation using chained equations to account for missing values [57].

For the primary outcome (RQ1), we will report point estimates and two-sided 95% CI. Non-inferiority will be concluded, and the null hypothesis rejected, if the lower bound of the CI exceeds the negative non-inferiority margin [58,59].

For secondary outcomes (RQ2, RQ3, and RQ4), we will report point estimates, two-sided 95% CI, and p-values, and the null hypothesis will be rejected if $p < 0.05$. RQ5 related outcomes will be reported descriptively using median and interquartile range.

We will adjust for multiple testing using the Holm method to control the family-wise error rate within each family of research questions, except for the primary outcome [58,60].

## Ethics and dissemination

This study is approved by the Danish Data Protection Agency (Privacy, P-2023–167), and the Danish National Committee on Health Research Ethics granted an exemption from requiring ethical approval (F-23005502). The study will adhere to the Declaration of Helsinki.

Participation is voluntary, and the principal investigator will collect written informed consent and permission to record and store video data for the research project from all participants upon enrolment (see supplementary material 1.) Participants may withdraw at any time without consequence, and the intervention will be discontinued if consent is revoked. Data will be stored securely in REDCap [54], with access limited to study investigators. Video recordings are stored on secure, access-restricted institutional servers, accessible only to authorized study personnel. All recordings are used solely for blinded performance assessment and are permanently deleted after completion of the analyses in accordance with the data management plan approved by the Danish Data Protection Agency.

The trial is registered on ClinicalTrials.gov, identifier: NCT06956833, https://clinicaltrials.gov/study/NCT06956833?term=NCT06956833&rank=1 on 04/24/2025. Any protocol changes will be updated on the registry. Results from the study will be submitted to peer-reviewed open-access journals and presented at national and international conferences. Participants who request it will receive a summary of the trial findings.

## Discussion

This study is expected to provide valuable insight into the effects of self-guided versus facilitator-guided debriefing in the context of immersive VR. The findings are expected to inform the future development and integration of VR into simulation-based medical education, expanding the understanding of both the potential advantages and limitations of self-guided debriefing.

Findings from our pilot study suggest that both debriefing methods are acceptable, usable, and effective for training interprofessional teams of medical doctors and nurses in a high resource setting [30]. We will add to these findings by investigating the two debriefing approaches in a population of more novice learners. In the absence of a facilitator, learners with less developed self-regulated learning skills may struggle to frame their experiences effectively [27,61], miss critical learning points, and experience a high cognitive load [27,28].

Further, our pilot study was based on a convenience sample. This study will add to our pilot findings by applying a non-inferiority RCT design, thus enabling us to investigate whether self-guided debriefing can be considered non-inferior to facilitator-guided debriefing for improving medical students' teamwork skills.

If self-guided debriefing proves comparably effective to facilitator-guided debriefing, it could enable greater flexibility in the timing, duration, and location of simulation-based medical education, potentially increasing access to simulation-based education in a global perspective. However, several barriers to implementing VR-based simulation remain, such as

logistical, cultural, and technological barriers [11,30,44,62]. Further, this trial involves medical students in a high-income country with access to advanced technology and little is known about the feasibility and effect of VR-based simulation in low resource settings [41,63,64]. Future studies could explore the feasibility and effect of VR in other contexts, particularly those with limited access to advanced technology, technical support, or prior simulation experience, to further inform for whom and under what conditions it is most effective.

### Strengths and limitations

Key strengths of this study include its randomized controlled design and the assessment of team performance by blinded raters using video recordings of participants. Further, we will employ a broad range of validated assessment tools, professionally translated to Danish and assessed as feasible for use in a Danish context. The preceding pilot study informed recruitment strategies, intervention delivery, and assessment tool feasibility, strengthening the trial's methodological foundation.

However, several limitations should be noted. First, the participant sample consists of medical students rather than residents or practicing clinicians, which may limit ecological validity and the direct applicability of findings to clinical practice. Second, the trial evaluates outcomes only in the short term, without long-term follow-up to assess retention of knowledge, maintenance of teamwork skills, or transfer of learning to real clinical environments. However, this design will help ensure a robust dataset with minimal missing data. Third, due to the inherent nature of comparing self-guided with facilitator-guided debriefing, participants cannot be blinded to group allocation, which may introduce performance bias. We mitigate this through standardized scenarios for both intervention delivery and performance evaluation, blinding of outcome assessors, use of validated assessment tools, and randomization to ensure balanced groups.

Fourth, randomization was performed at the team (cluster) level without stratification for potential confounding variables (e.g., prior VR or simulation experience), and team composition was not matched for, e.g., prior teamwork experience or clinical rotation level, which may introduce variability in baseline teamwork performance between groups. While baseline characteristics will be compared between groups, chance imbalances cannot be entirely excluded. Fifth, repeated testing introduces the risk of carryover or practice effects. Although scenarios differ in content but are matched for complexity, repeated exposure may lead to learning independent of the intervention. However, this is likely to affect both groups equally, potentially exaggerating overall improvements but not the primary non-inferiority comparison. Finally, although self-guided debriefing will be supported by standardized prompts to ensure consistency, there remains a possibility of variability in the depth and quality of reflection achieved between individuals and teams. These factors should be considered when interpreting the results.

### Supporting information

**S1 File. SPIRIT checklist 2025.**
(PDF)

**S2 File. Synopsis of protocol for ethical review inquiry (Danish Original).**
(PDF)

**S3 File. Synopsis of protocol for ethical review inquiry (English translation).**
(PDF)

### Acknowledgments

We gratefully acknowledge statistician Julie Forman for her valuable statistical advice and insightful consultation on selecting the appropriate statistical method for this study protocol.

## Author contributions

**Conceptualization:** Amalie Middelboe Sohlin, Anja Poulsen, Jette Led Sørensen, Jesper Kjærgaard.

**Funding acquisition:** Amalie Middelboe Sohlin, Anja Poulsen, Jette Led Sørensen, Jesper Kjærgaard.

**Methodology:** Amalie Middelboe Sohlin, Anja Poulsen, Ida Madeline Hoffmann, Line Klingen Gjærde, Stine Lund, Gritt Overbeck, Lone Paulsen, Todd P. Chang, Joy Yeonjoo Lee, Jette Led Sørensen, Jesper Kjærgaard.

**Project administration:** Amalie Middelboe Sohlin, Anja Poulsen, Jette Led Sørensen, Jesper Kjærgaard.

**Supervision:** Anja Poulsen, Jette Led Sørensen, Jesper Kjærgaard.

**Validation:** Anja Poulsen, Line Klingen Gjærde, Stine Lund, Gritt Overbeck, Todd P. Chang, Joy Yeonjoo Lee, Jette Led Sørensen, Jesper Kjærgaard.

**Writing – original draft:** Amalie Middelboe Sohlin.

**Writing – review & editing:** Amalie Middelboe Sohlin, Anja Poulsen, Ida Madeline Hoffmann, Line Klingen Gjærde, Stine Lund, Gritt Overbeck, Lone Paulsen, Todd P. Chang, Joy Yeonjoo Lee, Jette Led Sørensen, Jesper Kjærgaard.

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
