## [Decision Letter · Decision Letter 0]

6 Aug 2025

Dear Dr. Andersen,

Thank you for submitting your manuscript to PLOS ONE. After careful consideration, we feel that it has merit but does not fully meet PLOS ONE’s publication criteria as it currently stands. Therefore, we invite you to submit a revised version of the manuscript that addresses the points raised during the review process.

This study proposal is an interesting and important paper that compares the effects of self-guided compared to facilitator-guided debriefing for immersive virtual reality simulation-based pediatric emergency team training.

It is clear that time and energy have been dedicated to this study and overall, it is a well written study proposal.  I congratulate the research team for this excellent work. A few minor comments are given by the reviewers.

We look forward to receiving your revised manuscript.

Kind regards,

Ipek Gonullu, M.D., Ph.D.

Academic Editor

PLOS ONE

Journal Requirements:

“This work was supported by Gangstedfonden grant number A43424, Helsefonden grant number 21-B-385, and Laerdal Foundation grant number 2024-0321. The funders did not play any role in the study design, data collection and analysis, decision to publish, or preparation of the manuscript.”

“J.K. received a grant for this work from Gangstedfonden, grant number A43424, https://www.gangstedfonden.dk/ and Laerdal Foundation, grant number 2024-0321, https://laerdalfoundation.org/. A.P. received a grant for this work from Helsefonden, grant number 21-B-385, https://helsefonden.dk/ .

The funders did not play any role in the study design, data collection and analysis, decision to publish, or preparation of the manuscript.”

Additional Editor Comments (if provided):

This study proposal is an interesting and important paper that compares the effects of self-guided compared to facilitator-guided debriefing for immersive virtual reality simulation-based pediatric emergency team training.

It is clear that time and energy have been dedicated to this study and overall, it is a well written study proposal. I congratulate the research team for this excellent work. A few minor comments are given by the reviewers.

Reviewers' comments:

Reviewer's Responses to Questions

**Comments to the Author**

1. Does the manuscript provide a valid rationale for the proposed study, with clearly identified and justified research questions?

Reviewer #1: Yes

Reviewer #2: Yes

2. Is the protocol technically sound and planned in a manner that will lead to a meaningful outcome and allow testing the stated hypotheses?

Reviewer #1: Yes

Reviewer #2: Yes

3. Is the methodology feasible and described in sufficient detail to allow the work to be replicable?

Reviewer #1: Yes

Reviewer #2: Yes

4. Have the authors described where all data underlying the findings will be made available when the study is complete?

Reviewer #1: Yes

Reviewer #2: Yes

5. Is the manuscript presented in an intelligible fashion and written in standard English?

Reviewer #1: Yes

Reviewer #2: Yes

You may also provide optional suggestions and comments to authors that they might find helpful in planning their study.

Reviewer #1: Very minor revision

This is a well written protocol sensibly based on experience from an earlier randomised pilot study. The research questions are clearly specified (lines 153-168). The sample size appears appropriate although the description (lines 307-316) is not entirely clear. Could the authors look at the description again? The analytical methods with respect to the statistical analysis are entirely suitable.

Reviewer #2: The proposed study is timely and relevant, targeting a key gap in medical simulation pedagogy—whether self-guided debriefing can yield comparable learning outcomes to facilitator-led debriefing in immersive VR environments. The manuscript demonstrates strong methodological rigor, with a thoughtful use of pilot data to inform trial design and robust tools to evaluate both technical and non-technical outcomes.

A. Comments on methods’ section:

Strengths:

• The study is methodologically rigorous and adheres to the SPIRIT guidelines.

• It employs a well-defined randomized controlled, single-blinded, non-inferiority design, which is appropriate to answer the stated research questions.

• The use of validated and standardized assessment tools (e.g., CTS, ABCDE checklist, DASH-SV, IMI, NASA-TLX, SUS, VRSQ) strengthens the internal validity and reproducibility.

• Clear blinding procedures for outcome raters and detailed randomization methodology are commendable.

• The intervention and assessment phases are well-illustrated with timelines and scenarios.

• Considerations for VR-induced side effects (cybersickness) and ethical approvals are properly addressed.

Comments for Improvement:

1. Clarify rationale for non-inferiority margin:

o The chosen margin of 0.5 on the CTS scale appears derived from pilot data, but a clearer justification of its clinical relevance is warranted. The margin should reflect what constitutes a non-meaningful difference in teamwork skills.

2. Stratified randomization detail:

o Although stratified randomization is mentioned, the strata used (e.g., gender, prior VR/simulation experience) are not clearly specified. This should be detailed to ensure group comparability.

3. Team composition and matching:

o The formation of teams (pairs of students) may introduce variability. Consider clarifying whether any matching criteria (e.g., prior teamwork experience or clinical rotation level) were used to reduce heterogeneity across teams.

4. Handling missing data:

o The statistical plan notes an intention to account for missing data using mixed models, but further detail on handling attrition, particularly in individual-level data (e.g., imputation or exclusion), should be provided.

5. Blinding limitations:

o While rater blinding is described well, the lack of participant blinding may introduce performance bias. Although understandable, it could be useful to acknowledge this limitation more explicitly and discuss mitigation strategies.

6. Assessment of carryover effect:

o As the same individuals are involved in both pre- and post-tests with different scenarios, consideration of potential carryover learning effect or learning due to exposure (rather than intervention) is warranted.

7. Data safety and monitoring:

o The manuscript would benefit from a brief mention of any data monitoring committee or oversight mechanism, particularly in the context of student participants and recorded video data.

B. The Discussion and Limitations Sections:

Strengths:

• The discussion highlights the relevance and potential implications of scalable immersive virtual reality (VR) training, particularly its role in overcoming logistical and cost barriers in simulation-based medical education.

• The manuscript acknowledges the novelty of comparing self-guided versus facilitator-guided debriefing in immersive VR—a relatively unexplored area.

• The potential for broader implementation and democratization of simulation access is well-articulated and compelling.

Comments for Improvement:

1. Overly Optimistic Tone Without Data:

o The discussion leans heavily toward the anticipated benefits of self-guided debriefing without sufficient caution. Given this is a protocol paper, it would be more appropriate to moderate claims and clearly distinguish between expected outcomes and established findings from prior literature or pilot data.

2. Limited Integration of Pilot Study Findings:

o While pilot data are referenced earlier, their implications are not critically analyzed in the discussion. Incorporating a reflection on how these preliminary results informed both the study design and expected impact would enhance the rationale.

3. Theoretical Underpinnings Need Emphasis:

o The discussion would benefit from deeper integration of learning theory—particularly self-regulated learning and cognitive load theory—to better contextualize the study’s potential impact and risks (e.g., learner variability, overload).

4. Scalability Considerations Are Oversimplified:

o The promise of wider accessibility is emphasized, but logistical, cultural, and technological barriers to implementation (especially in low-resource settings) should be acknowledged.

5. Lack of Discussion on Generalizability:

o The study uses medical students in a high-income country with access to advanced technology. The discussion should more explicitly address the limitations in generalizing the findings to other populations, healthcare systems, or educational levels.

6. Limitations Section is Underdeveloped or Implied:

o A standalone "Limitations" paragraph or subsection is lacking. The discussion should explicitly list anticipated limitations such as:

Use of medical students rather than residents or clinicians (may affect ecological validity).

Short-term assessment only; no long-term follow-up to assess knowledge retention or skill transfer.

Inability to blind participants, which may introduce bias.

Potential variability in self-guided debriefing quality despite standardized prompts.

Overall, the manuscript is well-prepared and represents a valuable contribution to the field. Minor revisions as outlined above would strengthen clarity and transparency.

**Do you want your identity to be public for this peer review?** For information about this choice, including consent withdrawal, please see our Privacy Policy

Reviewer #1: No

Reviewer #2: No

---

## [Author Response · Author response to Decision Letter 1]

15 Aug 2025

RESPONSE TO EDITOR AND REVIEWERS

"Self-guided versus facilitator-guided debriefing in immersive virtual reality simulation: Protocol for a randomized controlled non-inferiority trial assessing teamwork skills in medical students" [PONE-D-25-26268]

EDITOR COMMENTS

We thank the Editor for the comment. We have updated the Supporting Information file names to comply with PLOS ONE’s style requirements and have ensured that the rest of the manuscript also adheres to these guidelines.

Please note that the first author’s last name has been changed, as I recently got married and adopted my married name.

“This work was supported by Gangstedfonden grant number A43424, Helsefonden grant number 21-B-385, and Laerdal Foundation grant number 2024-0321. The funders did not play any role in the study design, data collection and analysis, decision to publish, or prepa-ration of the manuscript.”

We note that you have provided additional information within the Acknowledgements Sec-tion that is not currently declared in your Funding Statement. Please note that funding in-formation should not appear in the Acknowledgments section or other areas of your manu-script. We will only publish funding information present in the Funding Statement section of the online submission form.

“J.K. received a grant for this work from Gangstedfonden, grant number A43424, https://www.gangstedfonden.dk/ and Laerdal Foundation, grant number 2024-0321, https://laerdalfoundation.org/. A.P. received a grant for this work from Helsefonden, grant number 21-B-385, https://helsefonden.dk/ .

The funders did not play any role in the study design, data collection and analysis, decision to publish, or preparation of the manuscript.”

We thank the Editor for this comment. We have removed the funding-related text from the manuscript and wish to retain the Funding Statement as currently written.

3. Please include captions for your Supporting Information files at the end of your manu-script, and update any in-text citations to match accordingly. Please see our Supporting In-formation guidelines for more information: http://journals.plos.org/plosone/s/supporting-information.

We thank the Editor for this comment and have now included captions for our Supporting Information files at the end our manuscript in line with the above guidelines.

4. If the reviewer comments include a recommendation to cite specific previously published works, please review and evaluate these publications to determine whether they are relevant and should be cited. There is no requirement to cite these works unless the editor has indi-cated otherwise.

Thank you for the comment. The reviewer comments did not include a recommendation to cite specific previously published works.

5. Please review your reference list to ensure that it is complete and correct. If you have cit-ed papers that have been retracted, please include the rationale for doing so in the manu-script text, or remove these references and replace them with relevant current references. Any changes to the reference list should be mentioned in the rebuttal letter that accompanies your revised manuscript. If you need to cite a retracted article, indicate the article’s retracted status in the References list and also include a citation and full reference for the retraction notice.

Thank you for the comment. We have reviewed our reference list to ensure it is complete and correct, and did not cite any papers that have been retracted. We have added the following references:

57. van Buuren S, Groothuis-Oudshoorn K. mice: Multivariate imputation by chained equations in R. J Stat Softw. 2011;45(3):1–67.

61. Panadero E. A review of self-regulated learning: Six models and four directions for research. Front Psychol. 2017;8(APR):1–28.

62. Savino S, Mormando G, Saia G, Da Dalt L, Chang TP, Bressan S. SIMPEDVR: using VR in teaching pediatric emergencies to undergraduate students—a pilot study. Eur J Pediatr [In-ternet]. 2024;183(1):499–502.

Additional Editor Comments (if provided):

This study proposal is an interesting and important paper that compares the effects of self-guided compared to facilitator-guided debriefing for immersive virtual reality simulation-based pediatric emergency team training.

It is clear that time and energy have been dedicated to this study and overall, it is a well written study proposal. I congratulate the research team for this excellent work. A few minor comments are given by the reviewers.

We thank the Editor for this valuable and encouraging feedback on our manuscript. We are grateful for your recognition of the strengths of our study proposal. We have carefully revised the manuscript in accordance with the reviewers’ comments. The specific revisions are described below.

We believe these revisions have enhanced the manuscript’s clarity and overall quality.

We look forward to your further consideration.

REVIEWER COMMENTS

Reviewer's Responses to Questions

Comments to the Author

1. Does the manuscript provide a valid rationale for the proposed study, with clearly identi-fied and justified research questions?

Reviewer #1: Yes

We thank the reviewer for confirming that the manuscript provides a valid rationale for the proposed study with clearly identified and justified research questions.

Reviewer #2: Yes

Comments: The manuscript provides a well-articulated rationale grounded in relevant litera-ture for evaluating self-guided versus facilitator-guided debriefing in immersive VR-based simulation. The identified research gap is significant, given the increasing adoption of VR in medical education and the need to assess the pedagogical equivalence of debriefing meth-ods. The research questions are clearly stated, aligned with the study objectives, and justi-fied based on the results of a preceding pilot study.

We thank the reviewer for the recognition of the significance of the identified research gap, the clarity of the research questions, and their alignment with the study objectives. We are also encouraged by your acknowledgement that the preceding pilot study supports the justification for the present trial.

2. Is the protocol technically sound and planned in a manner that will lead to a mean-ingful outcome and allow testing the stated hypotheses?

Reviewer #1: Yes

We thank the reviewer for confirming that the protocol is technically sound and well planned.

Reviewer #2: Yes

Comments: The study is designed as a randomized, controlled, non-inferiority trial with ap-propriate blinding of raters and a robust pretest-posttest structure. The use of validated as-sessment tools and statistical models (e.g., constrained linear mixed models) supports the technical rigor of the protocol. The inclusion of a power analysis based on pilot data adds strength to the proposed sample size. Exploratory outcomes, assumptions, and blinding are adequately addressed.

We thank the reviewer for acknowledging our efforts to ensure a robust study design, in-cluding the use of validated assessment tools, blinded raters, and power calculation based on pilot data.

3. Is the methodology feasible and described in sufficient detail to allow the work to be replicable?

Reviewer #1: Yes

We thank the reviewer for confirming that the methodology is feasible and described in sufficient detail.

Reviewer #2: Yes but

Comments:

Strengths:

• The study is methodologically rigorous and adheres to the SPIRIT guidelines.

• It employs a well-defined randomized controlled, single-blinded, non-inferiority design, which is appropriate to answer the stated research questions.

• The use of validated and standardized assessment tools (e.g., CTS, ABCDE checklist, DASH-SV, IMI, NASA-TLX, SUS, VRSQ) strengthens the internal validity and reproducibility.

• Clear blinding procedures for outcome raters and detailed randomization methodology are commendable.

• The intervention and assessment phases are well-illustrated with timelines and scenarios.

• Considerations for VR-induced side effects (cybersickness) and ethical ap-provals are properly addressed.

Comments for Improvement:

• Clarify rationale for non-inferiority margin: The chosen margin of 0.5 on the CTS scale appears derived from pilot data, but a clearer justification of its clinical relevance is war-ranted. The margin should reflect what constitutes a non-meaningful difference in teamwork skills.

• Stratified randomization detail: Although stratified randomization is mentioned, the strata used (e.g., gender, prior VR/simulation experience) are not clearly specified. This should be detailed to ensure group comparability.

• Team composition and matching: The formation of teams (pairs of students) may intro-duce variability. Consider clarifying whether any matching criteria (e.g., prior teamwork experience or clinical rotation level) were used to reduce heterogeneity across teams.

• Handling missing data: The statistical plan notes an intention to account for missing data using mixed models, but further detail on handling attrition, particularly in individual-level data (e.g., imputation or exclusion), should be provided.

• Blinding limitations: While rater blinding is described well, the lack of participant blinding may introduce performance bias. Although understandable, it could be useful to acknowledge this limitation more explicitly and discuss mitigation strategies.

• Assessment of carryover effect: As the same individuals are involved in both pre- and post-tests with different scenarios, consideration of potential carryover learning effect or learning due to exposure (rather than intervention) is warranted.

• Data safety and monitoring: The manuscript would benefit from a brief mention of any data monitoring committee or oversight mechanism, particularly in the context of student participants and recorded video data.

We thank the reviewer for this valuable and insightful feedback. Each of the above com-ments are addressed explicitly below in the section “6. Review Comments to the Author”.

4. Have the authors described where all data underlying the findings will be made available when the study is complete?

Reviewer #1: Yes

We thank the reviewer for confirming that we have described where data underlying the findings will be made available when the study is complete.

Reviewer #2: Yes

Comments: The authors have declared that no datasets were generated at this stage and that deidentified data will be made available upon completion of the study. Although acceptable, the statement could be slightly improved by identifying a planned repository or data-sharing platform for transparency (e.g., OSF, Dryad).

We thank the reviewer for this suggestion. While a specific data-sharing platform has not yet been selected, we will ensure that deidentified data will be available upon study com-pletion, in accordance with journal and institutional requirements.

5. Is the manuscript presented in an intelligible fashion and written in standard Eng-lish?

Reviewer #1: Yes

We thank the reviewer for recognizing that the manuscript is presented in an intelligible fashion.

Reviewer #2: Yes

Comments: The manuscript is clearly written, well-structured, and follows academic lan-guage conventions. Minor typographical or grammatical issues are negligible. The clarity and professionalism of writing make it easy to follow the study rationale and methodology.

We appreciate that the reviewer found the manuscript to be clearly written, well-structured, and easy to follow.

6. Review Comments to the Author

Please note that line numbers in the response to reviewers refer to the “Revised Manu-script with Track Changes” file.

Reviewer #1:

Very minor revision

This is a well written protocol sensibly based on experience from an earlier randomised pilot study. The research questions are clearly specified (lines 153-168). The sample size appears appropriate although the description (lines 307-316) is not entirely clear. Could the authors look at the description again? The analytical methods with respect to the statistical analysis are entirely suitable.

We thank the reviewer for this encouraging feedback on our manuscript and are pleased that the reviewer found the manuscript to be well written, the research questions clearly specified, and the analytical methods suitable.

We agree that the description of the sample size calculation needed clarification. We have revised the section (lines 311-319) to provide a clearer explanation of the calculation and justification for the non-inferiority margin.

Reviewer #2:

The proposed study is timely and relevant, targeting a key gap in medical simulation peda-gogy—whether self-guided debriefing can yield comparable learning outcomes to facilita-tor-led debriefing in immersive VR environments. The manuscript demonstrates strong methodological rigor, with a thoughtful use of pilot data to inform trial design and robust tools to evaluate both technical and non-technical outcomes.

A. Comments on methods’ section:

Strengths:

• The study is methodologically rigorous and adheres to the SPIRIT guidelines.

• It employs a well-defined randomized controlled, single-blinded, non-inferiority design, which is appropriate to answer the stated research questions.

• The use of validated and standardized assessment tools (e.g., CTS, ABCDE checklist, DASH-SV, IMI, NASA-TLX, SUS, VRSQ) strengthens the internal validity and reproduci-bility.

• Clear blinding procedures for outcome raters and detailed randomization methodology are commendable.

• The intervention and assessment phases are well-illustrated with timelines and scenarios.

• Considerations for VR-induced side effects (cybersickness) and ethical approvals are properly addressed.

We thank the reviewer for their thorough and encouraging assessment of our study proto-col. We appreciate their recognition of our efforts to ensure methodological rigor, adher-ence to SPIRIT guidelines, use validated assessment tools, maintain blinding of outcome raters, and address VR-induced side effects.

Comments for Improvement:

1. Clarify rationale for non-inferiority margin:

o The chosen margin of 0.5 on the CTS scale appears derived from pilot data, but a clearer justification of its clinical relevance is warranted. The margin should reflect what consti-tutes a non-meaningful difference in teamwork skills.

We thank the reviewer for this comment and agree that this part was not sufficiently clari-fied. We have now clarified the rationale for the chosen non-inferiority margin of 0.5 and why this margin is expected to constitute a non-meaningful difference in teamwork skills (lines 313-317):

“We set a non-inferiority margin of 0.5 points on the 0-10 CTS© scale, meaning that a dif-ference of less than 0.5 points on the CTS© scale is not expected to constitute a meaningful difference in teamwork skills. The non-inferiority margin was informed by expert consen-sus, our experience rating teamwork skills in the pilot study, and pilot data (30).”

2. Stratified randomization detail:

o Although stratified randomization is mentioned, the strata used (e.g., gender, prior VR/simulation experience) are not clearly specified. This should be detailed to ensure group comparability.

We thank the reviewer for pointing out this importan

---

## [Decision Letter · Decision Letter 1]

29 Aug 2025

Self-guided versus facilitator-guided debriefing in immersive virtual reality simulation: Protocol for a randomized controlled non-inferiority trial assessing teamwork skills in medical students

PONE-D-25-26268R1

Dear Dr. Andersen,

We’re pleased to inform you that your manuscript has been judged scientifically suitable for publication and will be formally accepted for publication once it meets all outstanding technical requirements.

Kind regards,

Ipek Gonullu, M.D., Ph.D.

Academic Editor

PLOS ONE

Additional Editor Comments (optional):

I thank the authors for addressing the reviewers’ comments.

Reviewers' comments:

Reviewer's Responses to Questions

**Comments to the Author**

1. Does the manuscript provide a valid rationale for the proposed study, with clearly identified and justified research questions?

Reviewer #1: Yes

2. Is the protocol technically sound and planned in a manner that will lead to a meaningful outcome and allow testing the stated hypotheses?

Reviewer #1: Yes

3. Is the methodology feasible and described in sufficient detail to allow the work to be replicable?

Reviewer #1: Yes

4. Have the authors described where all data underlying the findings will be made available when the study is complete?

Reviewer #1: Yes

5. Is the manuscript presented in an intelligible fashion and written in standard English?

Reviewer #1: Yes

You may also provide optional suggestions and comments to authors that they might find helpful in planning their study.

Reviewer #1: Accept

My earlier comment was very minor indeed. The small adjustment made adds to the clarity of the sample size calculation description.

**Do you want your identity to be public for this peer review?** For information about this choice, including consent withdrawal, please see our Privacy Policy

Reviewer #1: No

---

## [Editor Report · Acceptance letter]

PONE-D-25-26268R1

PLOS ONE

Dear Dr. Sohlin,

I'm pleased to inform you that your manuscript has been deemed suitable for publication in PLOS ONE. Congratulations! Your manuscript is now being handed over to our production team.

Kind regards,

on behalf of

Associate Professor Ipek Gonullu

Academic Editor

PLOS ONE